# The Impacts of Land Use Spatial Form Changes on Carbon Emissions in Qinghai–Tibet Plateau from 2000 to 2020: A Case Study of the Lhasa Metropolitan Area

Meimei Wang [1], Dezhen Kong [1], Jinhuang Mao [2], Weijing Ma [1,*] and Ramamoorthy Ayyamperumal [1]

1   College of Earth and Environmental Sciences, Lanzhou University, Lanzhou 730000, China
2   Institute of County Economic Development, Lanzhou University, Lanzhou 730000, China
*   Correspondence: maweijing@lzu.edu.cn

**Abstract:** The ecological contribution of the Qinghai–Tibet Plateau has received considerable attention as a result of the increased focus on global climate change and the continuous growth of carbon emissions in all countries. In this study, we proposed a method and measured the carbon emissions from land use in the Lhasa metropolitan area from 2000 to 2020, based on image interpretation data, by exploiting corrected carbon emission factors in different land types from the Qinghai–Tibet Plateau. We studied the impact of construction land form on carbon emissions using the spatial lag model (SLM) and the spatial error model (SEM), and the results show that the Lhasa metropolitan area's carbon emissions showed an overall increasing trend from 2000 to 2020, with the characteristics of "slow acceleration–slight deceleration–acceleration", with a deceleration period from 2005 to 2015. As a result, the construction land has a relatively low capacity, but it constitutes about 90% of all emissions; moreover, carbon emissions from cultivated land cover about 9%. The rate of spatial expansion of carbon emissions from land use is significantly slower in the Lhasa metropolitan area, yet the spatial expansion of carbon emissions has a clear direction and increases in the north and west of Lhasa. The carbon emissions from land use in the Lhasa metropolitan area is characterized by "one core, many points, and multiple belts" in spatial distribution. The changing of spatial forms of construction land has a significant impact on carbon emissions. Finally, we depicted the impact logic of land use pattern on carbon emissions and provided policy and management recommendations that were both feasible and reasonable.

**Keywords:** carbon emissions; land use form; influence mechanism; Lhasa metropolitan area; policy proposals

## 1. Introduction

Global climate change is now a big topic of concern for all countries around the world [1–3], and climate warming is one of the most serious environmental issues that China is currently facing. The global average temperature will eventually rise by 3–6 degrees Celsius if the world does not take urgent and aggressive action, which may cause sea levels to rise by tens of meters in the future. This catastrophe will lead to the extinction of almost half of the planet's species [4]. Carbon emissions are a key measure of greenhouse gas emissions, and an in-depth study of carbon emissions can help to understand the mechanisms by which they grow, and thus address environmental problems such as global warming [5]. Carbon emissions from land use change are the most uncertain part of the global carbon cycle [6,7], and different types of land use have an impact on carbon emissions due to their different land use intensities, human activities, and vegetation. For example, deforestation and other land cover changes typically release carbon from the terrestrial biosphere into the atmosphere in the form of $CO_2$ (carbon dioxide), which can lead to significant increases in regional carbon emissions. Moreover, land urbanization has a significant impact on carbon

emissions [8], and carbon emissions are the largest carbon source in the region, accounting for approximately 80% of the total regional carbon emissions.

Changes in the Chinese government's policies on land use are one of the main reasons for local land use changes and increases in the area of land for construction. The Chinese government required local governments to designate basic cultivated land reserves according to actual conditions, formulate village and township construction plans, and ensure that housing construction for rural residents conformed to village construction plans. To effectively protect cultivated land, the Chinese government prohibits the abandonment of cultivated land and attempts to separate land ownership and use rights to achieve land use in a paid, limited, and mobile manner. From 2000 to 2005, governments at all levels required the effective implementation of the strictest system of cultivated land protection and strict compliance with the authority and procedures for the approval of nonagricultural uses of cultivated land. In terms of land allocation, the Chinese government strictly controls the total supply of construction land, implements bidding and auctioning of state-owned land use rights, strengthens low price management, standardizes land approval, and introduces a reserve system for the acquisition of construction land. From 2005 to 2010, it pointed out the need to further define the responsibilities of principal officials of local governments at all levels for land management and cultivated land protection within their administrative areas. In forestry, the Chinese government has adjusted the plan of returning cultivated land to forest, realistically formulated the construction program of returning cultivated land to forest, and continued the reforestation of barren mountains. To strengthen the regulation and control of land allocation, and to play a supervisory role in construction land filling, the Chinese government once again emphasized the policy position of keeping cultivated land "unchanged for a long time", and further proposed to "extend the second round of land contracts for another 30 years after their expiry" from 2010 to 2020. In terms of land allocation, it is required to improve the control standards of land conservation and intensification, strengthen the assessment and evaluation of land conservation, promote land conservation models and technologies, and revitalize the stock of construction land.

Research on regional carbon emissions based on land use changes mainly focuses on four areas. The first is the accounting of $CO_2$-related revenue and expenditure for land use changes at various scales, as well as the study of their emission effects [8]. Researchers investigated at different scales, including national and local ones [9,10], such as using the carbon emission estimation model to estimate the carbon emission intensity and carbon footprint of different industrial spaces in China [11] and the carbon emission intensity and per capita land use carbon emissions of 30 national-level provinces in China from 2006 to 2016 [12]. Many researchers conducted studies on a regional scale [13–15], as well as studies on the vegetation carbon sink caused by land use changes from 1970 to 1990 and from 1990 to 2010 [16]; the changes in carbon emissions in a specific geographical area, including carbon emissions on the university campus of King Abdullah University of Science and Technology (KAUST); and the carbon flux changes caused by changes in tropical forests in the 1980s and 1990s [17]. The second research area is the different research methods of carbon emission drivers. Two decomposition methods are often used in academia: index decomposition analysis (IDA) and structural decomposition analysis (SDA) [18–21]. The driving forces of carbon emissions in northeast China were evaluated using the LMDI model [22], French energy was also analyzed using the LMDI method [23]. The third area is the influencing factors. Many researchers studied the impact of agricultural land transformation [24], the urban heat island effect [25], and urban households [6] on carbon emissions. Real GDP per capita, the urbanization rate, the ratio of tertiary–secondary industries, the renewable energy ratio, and fixed asset investment are used as key factors influencing China's carbon emissions [26]. The fourth area is on the impact of urban spatial structure on carbon emissions [27], such as finding that urban form influences internal carbon emissions and believing that it plays an important role in reducing carbon emissions [28,29] and the relationship between urban spatial structure

and carbon emissions in the Chinese context from 2002 to 2019 [30], where urban spatial agglomeration contributes to some extent to the reduction in carbon emissions [31].

Previous researchers analyzed the factors influencing carbon emissions, and Chinese academics focused on energy intensity at the national or provincial level, ignoring the impact of urban form on carbon emissions in urban areas of the Qinghai–Tibet Plateau. Tibet is the main body of the Qinghai–Tibet Plateau, the core area of the roof of the world, the water tower of Asia, and the third pole of the earth, as well as a crucial national barrier to ecological security. It is of great significance to study land use changes and carbon emission changes based on land use in the Lhasa metropolitan area in order to build an ecological Tibet, establish a barrier for ecological security, and maintain national ecological security. Land urbanization has a significant impact on carbon emissions [32,33], and carbon emissions are the largest carbon source in the region, accounting for approximately 80% of the total regional carbon emissions. The regions with intensive carbon emissions are metropolitan areas, which are important drivers of regional development, as well as important bodies responsible for China's efforts to achieve carbon peaking by 2030 and carbon neutrality by 2060. Metropolitan area development as well as land development and utilization are essential factors influencing regional carbon emissions. Therefore, it is urgent to better implement the new development concept to accelerate the low-carbon development of the plateau region at present. However, the analysis of land use changes and urban form in metropolitan areas and the study of their impact on carbon emission are crucial prerequisites for achieving this goal.

The Lhasa metropolitan area is the epitome of less-developed provinces and river valley cities in high mountain areas [34], and research on the carbon emissions from land use in the Lhasa metropolitan area and its impact mechanism can enrich the study of land-use carbon emissions in river valley cities in alpine regions. In addition, due to the weak economic development and infrastructure in the Qinghai–Tibetan Plateau, any damage to the ecosystem will result in much higher ecological management costs than in the central and eastern regions. Therefore, studying the carbon emissions and the impact mechanism of the plateau will be of great significance in building ecological security barriers in Tibet, especially in maintaining the national ecological security of China. In this research, we purposed data to analyze the land use changes in the Lhasa metropolitan area on the Qinghai–Tibet Plateau with the remote sensing data of land use changes, and introduced a spatial panel model by the coefficient estimation method to calculate the changes in carbon emissions from land use and reveal the influence of construction land form on regional carbon emissions. The research findings intended to provide a comprehensive picture of the current carbon emission and offer a scientific basis for the regional governments to formulate development and carbon emission reduction policies in the Qinghai–Tibet Plateau areas.

## 2. Data Source and Methods

### 2.1. Study Area

The Lhasa metropolitan area is affected by the plateau monsoon semiarid climate and subsidence airflow, with sunny weather throughout the year and over 3000 h of sunshine. Affected by high altitude, sunny weather, and underdeveloped economic conditions, the land use and carbon emissions of the Lhasa metropolitan area also show unique characteristics that are different from those in midwest valley cities and eastern plains cities in China. According to the level of economic development and population, we chose relatively developed urban areas on the Qinghai–Tibet Plateau for the study. The Lhasa metropolitan area in this paper covers 3 districts and 9 counties, that is, Chengguan District, Dagze District, Doilungdeqen District, Damxung County, Gongga County, Maizhokunggar County, Nedong County, Nyemo County, Quxu County, Sangri County, Linzhou County, and Ngagzha County (see Figure 1).

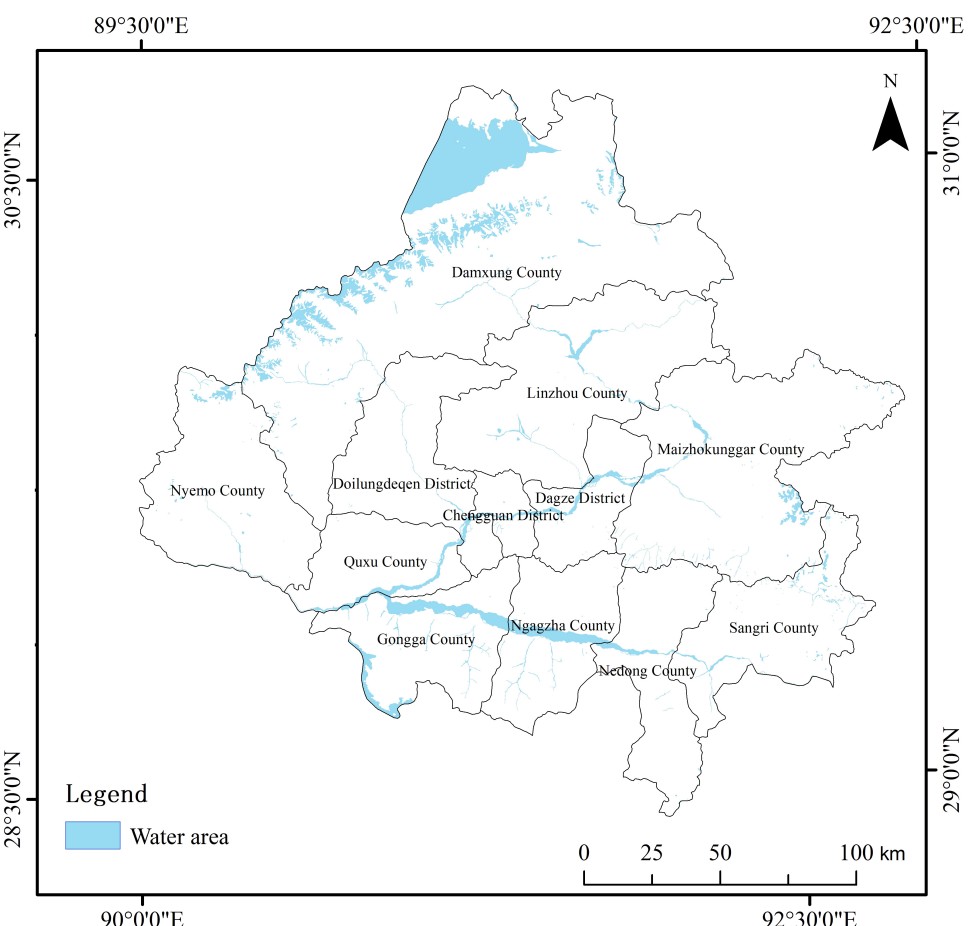

**Figure 1.** Study area.

*2.2. Data Sources*

The land data of Lhasa metropolitan area in this study came from the Science data Center of Resources and Environment of the Chinese Academy of Sciences (http://www.resdc.cn accessed on 1 June 2021.) for five periods from 2000 to 2020. The land is classified into 6 types, namely forest land, grassland, cultivated land, water space, construction land and unused land; the accuracy is 30 m.

*2.3. Methods*

(1)　Comprehensive measurement method

Carbon sources in land use have a positive value and carbon sinks have a negative value. The carbon emissions for different land types are calculated by the following equation:

$$E = \sum e_i = \sum S_i \times T_i \tag{1}$$

where E is total carbon emissions; $e_i$ is carbon emissions from an individual land type; $S_i$ is land area corresponding to an individual land type; $\delta_i$ is carbon emission factor per unit area of an individual land type, with i representing 6 different land types, respectively.

(2)　Spatial panel data model

The spatial lag model (SLM) and spatial error model (SEM) are introduced for the analysis of carbon emissions in the Lhasa region to investigate the influence of construction land form on regional carbon emissions. The maximum likelihood method is used to estimate the parameters of SLM and SEM.

 (1) Spatial lag model (SLM) is usually used to investigate the presence of diffusion of variables in the region (spillover effects). The formula is:

$$Y = \rho WY + X\beta + \varepsilon \tag{2}$$

where Y is the dependent variable, i.e., the degree of diffusion; X is the matrix of exogenous explanatory variables at order n × k; ρ is the spatial regression coefficient; w is the matrix of spatial weights at order n × n; WY is the spatial lagged dependent variable; the parameter β reflects the effect of the independent variable X on the dependent variable Y; and ε is the vector of random error terms.

(2)     Spatial Error Model (SEM)

The mathematical expression for the spatial error model (SEM) is:

$$y = X\beta + \varepsilon; \varepsilon = \lambda W\varepsilon + \mu \tag{3}$$

where ε is the vector of random error terms and λ is the spatial error coefficient of the vector of cross-sectional dependent variables at order n × 1. The parameter β reflects the effect of the independent variable X on the dependent variable y. The parameter λ measures the role of spatial dependence in the sample observations.

*2.4. Index Selection*

The influencing factors of eight landscape indexes, namely area-weighted mean patch fractal dimension (AWMPFD), largest patch index (LPI), mean perimeter–area ratio (CONTIG_MN), cohesion index (COHESION), aggregation index (AI), plaque adjacency (PLADJ), landscape shape index (LSI), and mean contiguity index (CONTIG MN), were constructed and used to analyze their influence based on the local construction land area. The above indicators, as shown in Table 1, represent the complexity of the urban landscape, boundary, fragmentation, dominant patches, connectivity, and overall compactness.

**Table 1.** Impact factors based on land-use carbon emissions.

| Index | Abbreviation | Index Significance |
|---|---|---|
| Largest patch index | LPI | The total length of all patch boundaries in all landscape type |
| Area-weighted mean patch fractal dimension | AWMPFD | The complexity of the spatial shape of patches and landscapes |
| Mean perimeter–area ratio | PARA_MN | The complexity of the shape of land use |
| Mean contiguity index | CONTIG_MN | The dominance of landscape |
| Plaque adjacency | PLADJ | Degree of aggregation of land use types |
| Cohesion index | COHESION | Aggregation and dispersion of patches in landscape |
| Aggregation index | AI | Connectivity between landscape types of patches |
| Landscape shape index | LSI | The total length of all patch boundaries of landscape types |

*2.5. Impact Logic of Land Use Form on Carbon Emission*

Land use changes in Lhasa metropolitan area are based on policy changes by the central and local governments in China, and the land use changes are directly responsible for the increase in carbon emissions. However, the real microscopic factor that contributes to changes in carbon emissions is the change in construction land, especially its spatial patterns. Therefore, based on policy changes, land use changes, and changes in the spatial patterns of construction land, this paper will propose a logical hypothesis that the internal compactness, complexity, and connectivity of construction land may affect the changes in carbon emissions. Local policies, land development, and the urban form and changes in it resulting from changes in the first two elements are the core factors leading to significant changes in local carbon sources, and thus in total carbon emissions. The proposal and adjustment of policies and the design and implementation of planning may lead to certain changes in local land use; affected by the development and transformation of different types of land, construction land will develop in different patterns. These patterns and their changes are the core microscopic factors that lead to a significant increase or decrease in local carbon sources. The logic hypothesis diagram is shown in Figure 2.

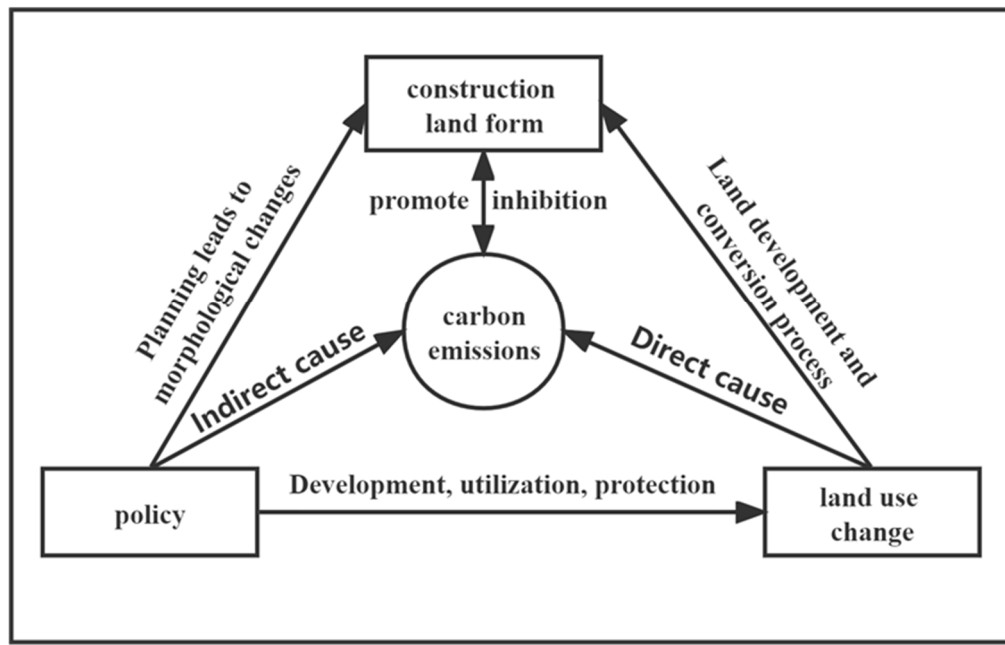

**Figure 2.** Logic hypothesis diagram.

In general, land use changes in the Lhasa metropolitan area are influenced by policy changes of the central and local governments and are directly responsible for changes in local carbon sources and carbon sinks. Regional carbon emissions are directly affected by the area of construction land. The shapes of the landscape, the complexity of boundaries, and the degree of fragmentation of construction land are all positively correlated with carbon emissions at the microscale, while construction land compactness is negatively related to carbon emissions. The impact of dominant patches and construction land connectivity on carbon emissions will vary depending on land use patterns.

## 3. Results and Analysis

### 3.1. Spatiotemporal Evolution Characteristics of Land Use

According to the land use data interpreted by remote sensing, the Lhasa metropolitan area covered a total land area of 38,956.63 km$^2$ in 2020. Specifically, cultivated land covered 1061.67 km$^2$, accounting for 2.72% of the total land area; construction land covered 74.17 km$^2$, accounting for 0.19%; grassland covered 27,530.40 km$^2$, accounting for 70.66%; forest land covered 2017.89 km$^2$, accounting for 5.17%; water space covered 2143.72 km$^2$, accounting for 5.50%; and unused land covered 6128.78 km$^2$, accounting for 15.73%. The land use in the Lhasa metropolitan area was characterized by (1) the highest proportion of grassland, followed by unused land and water, and the lowest proportion of construction land, indicating a good ecological foundation and good ecological and environmental functions in the Lhasa metropolitan area; (2) a low development intensity, with construction land accounting for only 0.19%, much lower than that of the Xining Metropolitan Area, another metropolitan area on the Qinghai–Tibet Plateau, indicating a low intensity of human activities in the Lhasa metropolitan area, with a low level of urbanization of the metropolitan area and a small scale of cities (Figure 3).

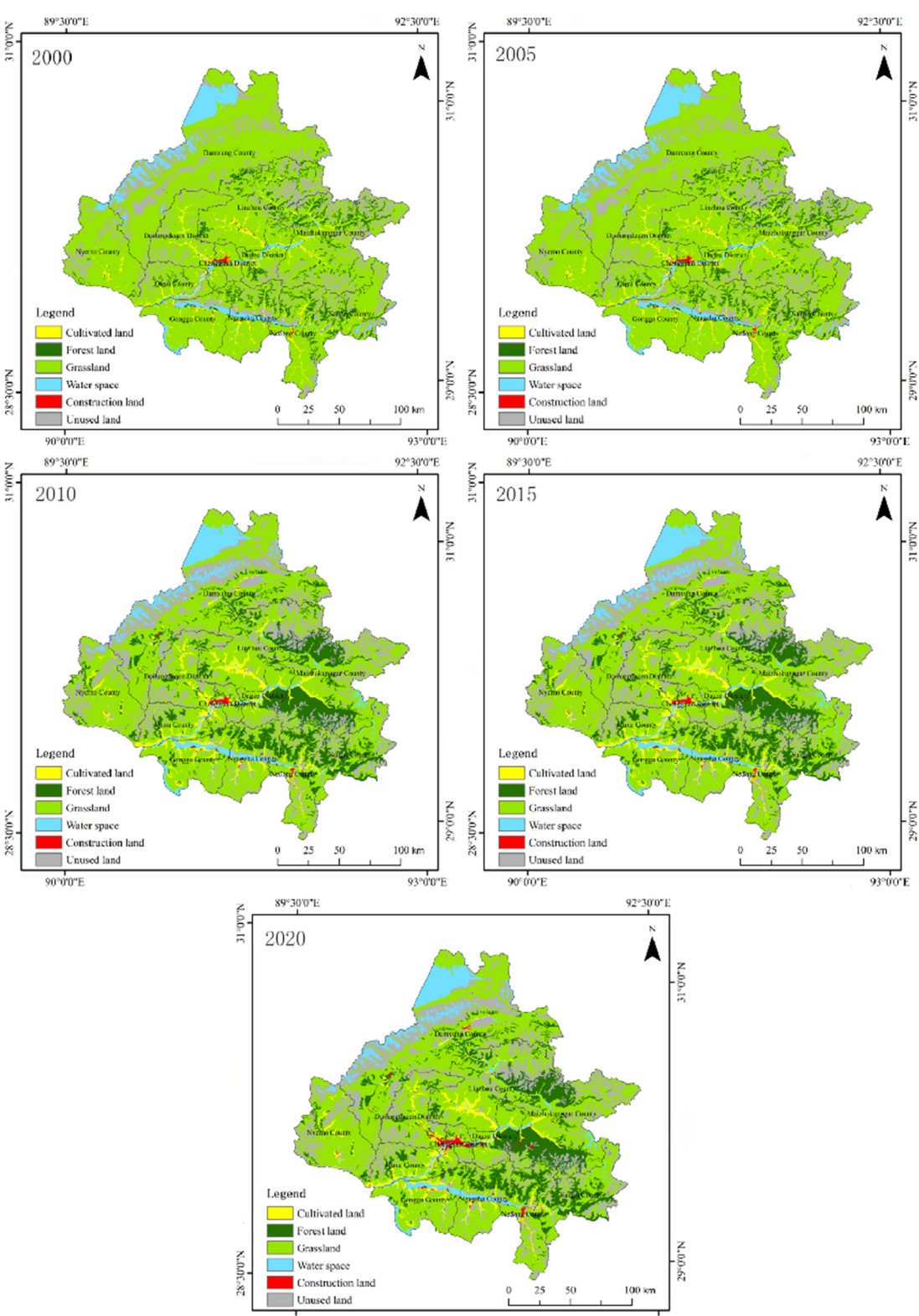

**Figure 3.** Land use data of Lhasa metropolitan area in different periods.

Comparing the growth rates of various types from 2000 to 2020, the results are shown in Table 2. Firstly, the grassland in the Lhasa metropolitan area was the most widely distributed, continuous, and in deceleration over the last 20 years, with a total decrease of 5816.96 km$^2$; the forest land showed "acceleration–deceleration–acceleration" changes with the largest increase by 3123.63 km$^2$; the water and unused land changed a little,

characterized by "deceleration–acceleration–deceleration". Secondly, the construction land was characterized by "one core with many points", where the term "one core" means the construction land concentrated in Lhasa's main urban area, and the term "many points" means the construction land is spot-like or lumped in other downtown areas. This type of land use showed an "acceleration–deceleration–acceleration" growth with relatively drastic changes in the 20 years; specifically, it decelerated from 2005 to 2010 and accelerated from 2010 to 2020. It increased by 142.11 km$^2$ from 2015 to 2020, up to 167.93%, indicating that the Lhasa metropolitan area experienced a rapid urbanization and industrialization process from 2015 to 2020. Compared with the cities in the eastern plains, the cities in the study area showed no significant changes in the spatial expansion of construction land, and the expansion of construction land is concentrated in the northern and western regions of Lhasa due to its long-term economic backwardness and harsh natural conditions, as well as the limited urban development in the Lhasa metropolitan area. Thirdly, the cultivated land in the Lhasa metropolitan area was distributed along the river valley and was concentrated in the valleys of the Yarlung Zangbo River, including the tributaries of the Lhasa River, the Nianchu River, the Niyang River, the Pulong Zangbo River, and some second-level tributaries. The cultivated land presented the change characteristics of "deceleration–acceleration–deceleration", stable with a little change overall, due to the implementation of farmland protection policy across the country.

**Table 2.** Land use change in Lhasa metropolitan area.

| Index | Period | Cultivated Land | Forest Land | Grassland | Water Space | Construction Land | Unused Land |
|---|---|---|---|---|---|---|---|
| Variation (km$^2$) | 2000–2005 | −7.59 | 1.58 | −7.98 | −0.44 | 14.93 | −0.51 |
| | 2005–2010 | 475.66 | 3152.68 | −5778.01 | 369.56 | −6.16 | 1786.28 |
| | 2010–2015 | −1.12 | 0.15 | −0.51 | −0.26 | 1.69 | 0.05 |
| | 2015–2020 | −93.54 | −30.78 | −30.45 | 19.51 | 142.11 | −6.84 |
| | 2000–2020 | 373.41 | 3123.63 | −5816.96 | 388.37 | 152.57 | 1778.97 |
| Rate of change (%) | 2000–2005 | −0.71 | 0.08 | −0.03 | −0.02 | 20.13 | −0.01 |
| | 2005–2010 | 45.13 | 156.11 | −20.99 | 17.24 | −6.92 | 29.15 |
| | 2010–2015 | −0.07 | 0.00 | 0.00 | −0.01 | 2.04 | 0.00 |
| | 2015–2020 | −6.12 | −0.60 | −0.14 | 0.78 | 167.93 | −0.09 |
| | 2000–2020 | 35.17 | 154.80 | −21.13 | 18.12 | 205.71 | 29.03 |

### 3.2. Spatiotemporal Evolution Characteristics of Carbon Emissions on Land Use

The net land-use carbon emission factors in this paper are mainly collected from the results of previous studies, and they are corrected and calculated. According to the current research results, the net carbon emission factor for cultivated land is 0.0497 kg/(m$^2$·a) [35], and the carbon emission factor for grassland is −0.0021 kg/(m$^2$·a) [36]. Carbon emissions from water space are mainly the emissions or sequestration of carbon by wetlands [37]. In this paper, the average carbon sink factor of water space for carbon emissions from land use is 0.0257 kg/(m$^2$·a) [38], and the final value is −0.0253 kg/(m$^2$·a), calculated by the average carbon emission factor of water space. Due to the complexity of human activities, it is difficult to calculate the carbon emission of construction land directly, and they are generally estimated indirectly by the energy consumption in the human activities and the carbon emission factor of each energy source. Based on the calculation results of previous research, the average carbon emission intensity for construction land is 3.607 kg/(m$^2$·a) [35]. Unused land mainly refers to barren hills, sandy land, bare rocky gravel land, bare land, barren grassland, and other types that are difficult to develop, with low biomass and low carbon emissions and carbon sequestration. According to existing articles, the carbon emission factor of unused land is −0.0005 kg/(m$^2$·a).

Based on the carbon emission data of the Lhasa metropolitan area, the following fascinating phenomena can be found using the determined emission factors for calculation:

(1) From 2000 to 2020, the total carbon emissions on land use in the Lhasa metropolitan area showed an overall increasing trend, characterized by "slow acceleration–slight deceleration–acceleration", with a deceleration period from 2005 to 2015, specifically 193,457.90 t, 246,952.35 t, 231,947.10 t, 237,994.02 t, and 745,688.45 t for 2000, 2005, 2010, 2015, and 2020, respectively. Compared with cities in the eastern plains, cities in the Lhasa metropolitan area had lower carbon emissions of land use [36], much smaller than those in the Xining metropolitan area.

(2) The overall carbon source–sink ratio remained constant from 2000 to 2020, only increasing sharply from 2.66 to 6.2 from 2015 to 2020, specifically 2.53, 2.95, 2.62, 2.66 and 6.20 for 2000, 2005, 2010, 2015 and 2020, respectively, which was associated with the sharp increase in construction land from 2015 to 2020.

(3) Construction land accounted for the smallest part, but played a significant role in the growth of carbon emissions. From 2000 to 2010, construction land accounted for a decreased proportion of 79.7% from 83.5% in total carbon sources, and then an increased proportion to 91.9% in 2020 (Figure 4). Compared with eastern plain cities, the proportion of carbon emissions from construction land in the Lhasa metropolitan area is relatively smaller, for the proportion of carbon emissions from construction land in eastern plain cities in China is about 95% [37].

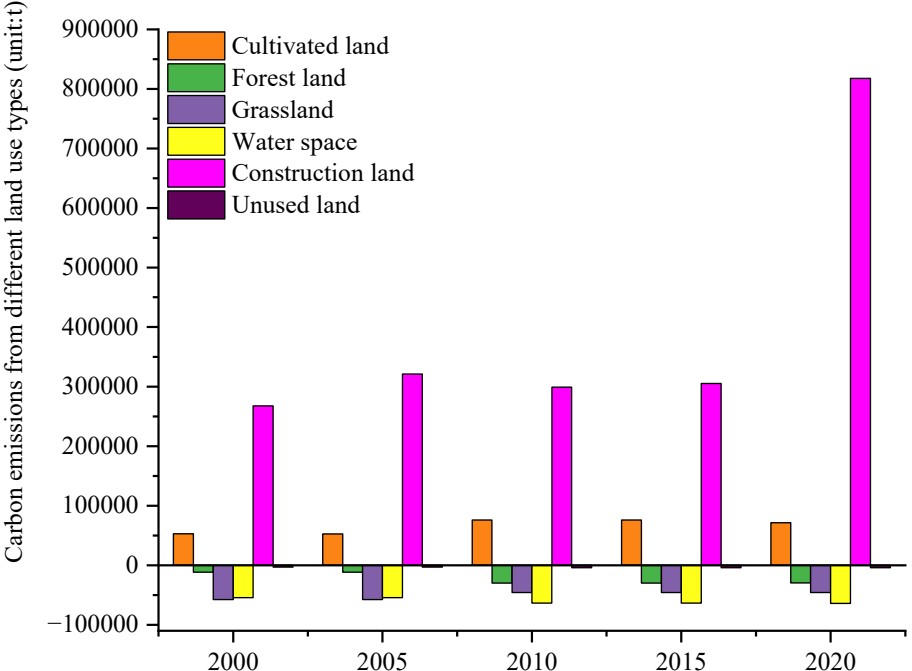

**Figure 4.** The carbon emissions of different land use types in Lhasa metropolitan area.

(4) Cultivated land accounted for about 9% of carbon emissions, growing slowly with a little change over the past 20 years. Forest land, grassland, water space, and unused land were carbon sinks. From 2000 to 2020, changes in carbon sequestration capacity due to changes in forest land, grassland, water space, and unused lands in the metropolitan area were weak, with a little change. (Figure 4)

Based on the spatial distribution of carbon emissions from land use in the Lhasa metropolitan area (Figure 5), it is found that:

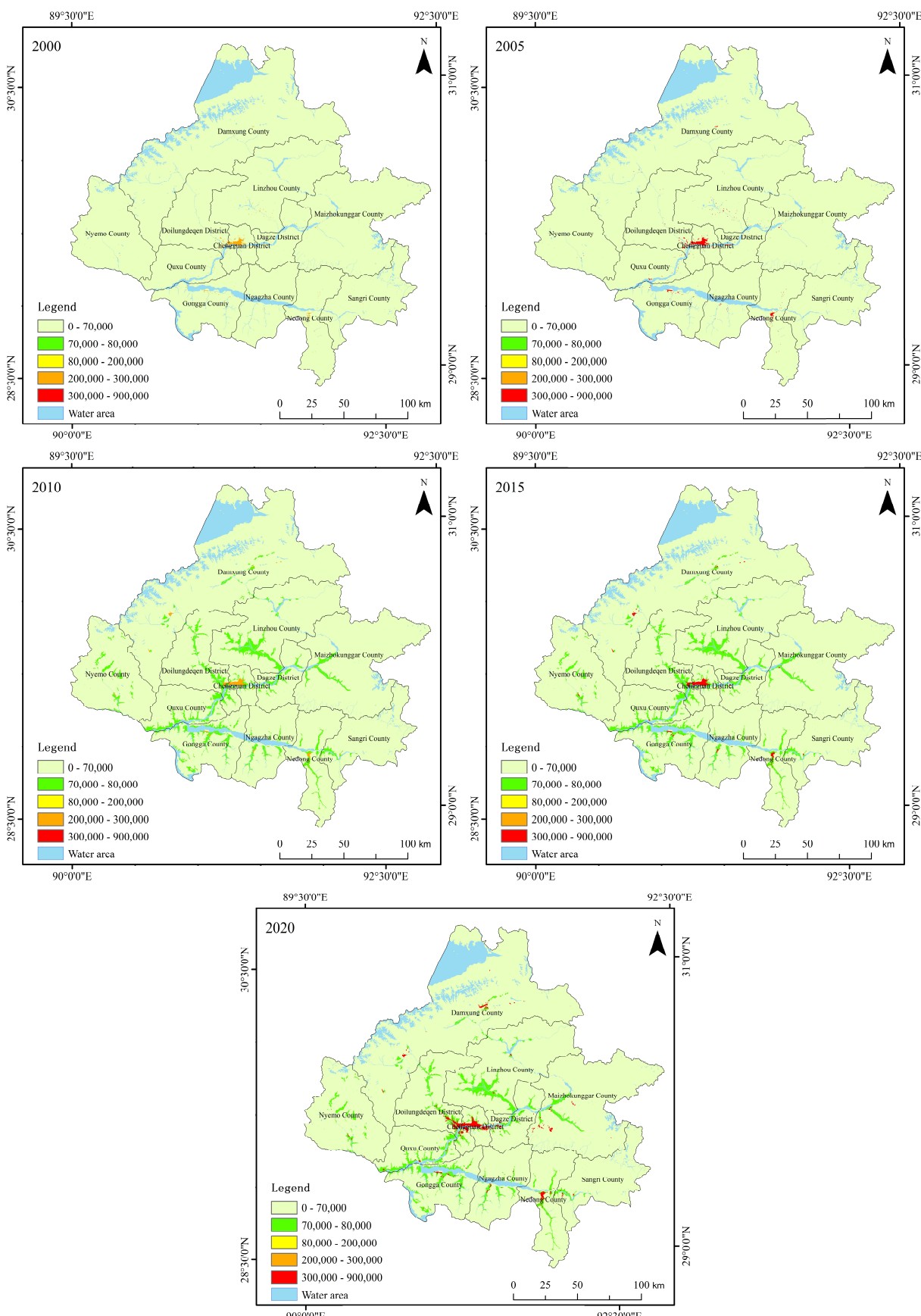

**Figure 5.** Carbon emissions in Lhasa metropolitan area from 2000 to 2020.

(1)　The carbon emissions from land use in the Lhasa metropolitan area were characterized by "one core, many points, and multiple belts" in spatial distribution, where the term "one core" means that carbon emissions were relatively concentrated in the main urban area of Lhasa City, the term "many points" means that carbon emissions were in the form of dots or clusters in the rest of the Lhasa metropolitan area, and the term "multiple belts" means that there were slight bands of carbon emissions in the vast suburbs of the Lhasa metropolitan area.

(2)　The carbon emission concentration of construction land was relatively high, mainly concentrated in the main urban area of Lhasa City and downtown of the rest of the Lhasa metropolitan area, forming "one core and multiple points". The low concentration of carbon emissions from cultivated land and the scattered spatial distribution contributed to the formation of "multiple belts" in the tributaries of the valleys of the Yarlung Zangbo River.

(3)　Carbon emissions concentrated in the main urban area of Lhasa City presented a circle-layer epitaxy expansion pattern, and there were differences in the expansion patterns at different stages. From 2000 to 2005, it showed a circle-layer type of epitaxy expansion model, and showed an axial type from 2005 to 2010. After 2010, it showed a change from epitaxial expansion to circle-layer epitaxial expansion, and the carbon emissions in the peripheral areas of Chengguan District also increased, then gradually formed a continuous distribution with the main urban area.

(4)　Constrained by the harsh natural conditions and backward economic conditions, the spatial expansion rate of carbon emissions from land use in the Lhasa metropolitan area was much lower than that of the eastern plain, with fewer new urban areas in the main city and the increase in new urban areas concentrated in the north and west of Lhasa. Thus, the spatial expansion of carbon emissions was concentrated in the north and west of Lhasa. in addition, carbon emissions from downtown land use in other areas of the Lhasa metropolitan area were scattered and clustered, with widely varying growth rates.

The spatial distribution of carbon emissions from land use in the Lhasa metropolitan area is different from that in the eastern cities. Carbon emissions in Cities in eastern China, such as the Pearl River Delta, have a centralized carbon emission core, which is larger in scope and gradually expands outward from the periphery [29]. The construction land and agricultural land in the Lhasa metropolitan area form a carbon emission pattern of "one core and multiple belts" with a very small carbon emission core area. The "multibelt" carbon emission distribution formed by agricultural land is due to the lack of heat in the Qinghai–Tibet region and the relatively low annual effective temperature, which cannot meet the maturation requirements of general crops. Therefore, in order to solve the problem of heat deficiency, the plantation industry in the Qinghai–Tibet region is concentrated in the valleys of the Yarlung Tsangpo River and Huangshui River and its tributaries, forming a unique "one core and multiple belts" carbon emission pattern.

## 4. Relationship between Land Form and Carbon Emissions

The correlation analysis results show that changing the land form of forest land, grassland, water space, unused land, and cultivation land has no significant effect on carbon emissions and carbon source/sink, whereas the changing of construction land has a significant effect on carbon emissions (Figure 6). Then, we will analyze the impact of construction land form on carbon emissions of land use.

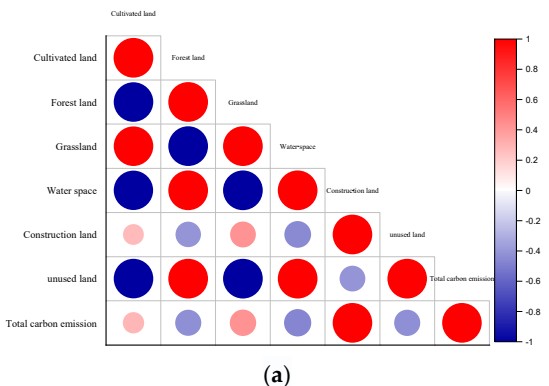

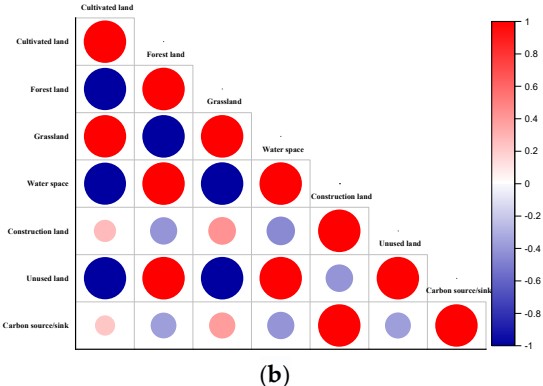

**Figure 6.** Correlation between land types and total carbon emissions, carbon source/sink. (**a**) Correlation between land types and total carbon emission; (**b**) Correlation between land types and carbon source/sink.

*Impacts of Landscape Indicators on Carbon Emission*

The results of SLM and SEM show that the relevant influencing factors were identified using panel data for a total of five time periods ranging from 2000 to 2020 (Tables 3 and 4). The estimated coefficient values and the significance level test show that SLM and SEM have a similar fit and the analysis results are similar. With the development of the economy, the influence of construction land form on carbon emission diminished.

**Table 3.** Impacts of landscape indicators on carbon emission of construction land on SLM.

|  | 2000 | | 2005 | | 2010 | | 2015 | | 2020 | |
|---|---|---|---|---|---|---|---|---|---|---|
|  | β | *p* | β | *p* | β | *p* | β | *p* | β | *p* |
| LPI | −37.446 | *** | −79.593 | *** | −79.732 | ** | 12.077 | 0.794 | −83.931 | *** |
| LSI | 320.116 | *** | 530.971 | *** | 4002.900 | *** | 3122.880 | *** | 223.314 | 0.713 |
| AWMPFD | 177,021.000 | *** | 318,076.000 | *** | 404,350.000 | *** | 389,603.000 | *** | 552,212.000 | *** |
| PARA_MN | −132.595 | *** | 312.421 | *** | −612.833 | * | −18.652 | 0.916 | 452.908 | *** |
| CONTIG_MN | −165,884.000 | *** | 370,503.000 | *** | −747,018.000 | * | −17,510.900 | 0.937 | 473,056.000 | *** |
| PLADJ | −602.206 | *** | −3652.740 | *** | −8590.990 | *** | −6566.680 | *** | −1815.390 | 0.525 |
| COHESION | −3027.260 | *** | −4907.510 | *** | −13,175.100 | *** | −12,951.600 | *** | −14,071.300 | *** |
| AI | 3258.880 | *** | 7704.110 | *** | 20,319.800 | *** | 16,253.700 | *** | 12,319.800 | *** |

Note: ***, **, and * represent significant levels of 1%, 5%, and 10%, respectively.

**Table 4.** Impacts of landscape indicators on carbon emissions of construction land on SEM.

|  | 2000 | | 2005 | | 2010 | | 2015 | | 2020 | |
|---|---|---|---|---|---|---|---|---|---|---|
|  | β | *p* | β | *p* | β | *p* | β | *p* | β | *p* |
| LPI | −33.368 | *** | −106.723 | *** | −141.186 | * | −24.935 | 0.637 | −96.451 | 0.111 |
| LSI | 373.394 | *** | 9.004 | 0.965 | 2592.860 | ** | 2906.250 | *** | 254.187 | 0.901 |
| AWMPFD | 175,093.000 | *** | 312,108.000 | *** | 394,192.000 | *** | 375,301.000 | *** | 434,359.000 | *** |
| PARA_MN | −116.719 | *** | 103.877 | 0.207 | −531.666 | 0.195 | −2.073 | 0.991 | 271.604 | 0.437 |
| CONTIG_MN | −147,546.000 | *** | 120,848.000 | 0.246 | −655,704.000 | 0.186 | −1305.720 | 0.996 | 261,181.000 | 0.522 |
| PLADJ | −804.368 | *** | −1792.450 | *** | −7000.540 | *** | −6339.960 | *** | −2641.400 | 0.792 |
| COHESION | −3007.350 | *** | −4850.490 | *** | −10,519.900 | *** | −11,673.000 | *** | −11,000.500 | ** |
| AI | 3465.110 | *** | 5549.280 | *** | 17,196.700 | *** | 15,624.100 | *** | 12,042.100 | 0.380 |

Note: ***, **, and * represent significant levels of 1%, 5%, and 10%, respectively.

Based on the two models, it is considered that the landscape shape index (LSI), area-weighted mean patch fractal dimension (AWMPFD), and aggregation index (AI) are the principal factors promoting carbon emissions from construction land in the Xining metropolitan area. In the spatial error model, the five indexes LPI (largest patch index), PLADJ (plaque adjacency), CONTIG_MN (mean contiguity index), PARA_MN (mean perimeter-area ratio), and COHESION (cohesion index) show significant negative correlations, while in the results of the spatial lag model-based analysis, the PARA_MN and CONTIG_MN correlations changed from negative to positive from 2000 to 2005.

Firstly, LSI and AWMPFD are important indexes for measuring urban complexity. The total length of the boundary of all patches in the landscape type is represented by LSI, and its value is larger when the shape of patches in the landscape deviates more from the square. Based on fractal theory, AWMPFD is used to calculate the spatial shape complexity of patches and landscapes. These two indexes are positively correlated with carbon emissions, indicating that the more complex landscape and boundary shape of construction land in the Lhasa metropolitan area result in higher carbon emissions. The reason for this is that regional economic development is accompanied by further development and utilization of construction land, particularly in the Lhasa metropolitan area, where the small area covered by construction land, low level of development, and high cost of expanding outward lead to continuous development and energy consumption of the original construction land, thereby increasing carbon emissions. It is also demonstrated by a shift in the CONTIG_MN correlation, where CONTIG_MN is the mean perimeter–area ratio and a higher value indicates a more complex shape.

Secondly, AI is the aggregation index, which reveals the connectivity of the landscape, whereas CONTIG_MN represents the degree of landscape fragmentation. In general, a high mean contiguity index indicates a high degree of landscape fragmentation. It suggests that a higher degree of construction land connectivity contributes to a higher degree of fragmentation and land concentration in the Lhasa metropolitan area, and thus leads to an increase in carbon emissions.

Thirdly, there is a significant negative correlation between LPI, PLADJ, and COHESION. PLADJ is a specific type of measure of aggregation, representing the aggregation degree of patches in a class of landscapes; COHESION also reflects the aggregation and dispersion of patches in the landscape. These two indicators reflect the degree of urban compactness, indicating that the more compactness of the construction land internally leads to more significant suppression of carbon emission. LPI is equal to the proportion of the largest patch of a given type occupying the entire landscape area, which is helpful to determine the modal or dominant type of the landscape. LPI shows a certain negative correlation, but there is no significant effect, because in the Qinghai–Tibet Plateau region, the Lhasa metropolitan area covers a large area with a small proportion of construction land, determining that the clustering and expansion of certain types of buildings within the construction land to a certain extent will not lead to the growth of regional carbon emissions.

## 5. Conclusions and Policy Recommendations

### 5.1. Conclusions

We attempted to provide a comprehensive accounting of carbon emissions in the study, and investigated the impact of construction land form on carbon emissions using the spatial lag model (SLM) and the spatial error model (SEM). Our findings are as follows:

(1) From 2000 to 2020, the carbon emissions of the Lhasa metropolitan area were generally on the rise, showing the characteristics of "slow acceleration–slight deceleration–acceleration", with a deceleration period from 2005 to 2015. Construction land accounted for the smallest proportion, but contributed to about 90% of carbon emissions, compared to cultivated land producing 9% of carbon emissions. Carbon emissions from land use in the Lhasa metropolitan area were characterized by "one core, multiple points and multiple belts" in the spatial distribution. Compared with other metropolitan areas, the Lhasa area was significantly slower in the spatial expansion of carbon emissions from land use. However, the spatial expansion of carbon emissions showed a clear direction and increased in the north and west of Lhasa.

(2) With the development of economy, the influence of construction land form on carbon emission was weakening. The results of the spatial lag model (SLM) and spatial error model (SEM) showed that while changing the construction land had a considerable effect on carbon emissions, changing the land for cultivation had almost no effect.

(3) The amount of land used for construction had a direct impact on local carbon emissions. At the microscale, there was a positive correlation between the morphology

of the landscape, the complexity of its boundaries, and the degree of land fragmentation used for development, while the amount of construction land was negatively correlated with carbon emissions.

*5.2. Policy Recommendations*

China is in the process of accelerated industrialization and urbanization, with a strong demand for land. Due to differences in natural conditions, economic and social development status, and development stages, different regions face different land use problems in the development process, presenting different situations and characteristics. To achieve sustainable development and put the "double carbon plan" into effect in the metropolitan area of Lhasa, it is important to promote the coordination of regional land use and carbon emission reduction from a holistic perspective. The local government still prefers traditional manners such as "afforestation" and "reforestation" to reduce carbon emissions, i.e., to increase carbon absorption. According to this paper, changing land use patterns will inhibit the growth of carbon sources and thus reduce local carbon emissions. As a result, we propose the following feasible and reasonable policy and management recommendations for the current land use and carbon emissions in the Lhasa metropolitan area:

(1)    Differentiated land use policies should be adopted.

The Lhasa metropolitan area should manage its land use in accordance with the *National Land Use Planning Outline* (2016–2030) and act according to the actual circumstances. In view of significant differences within the land use area of 38,956.63 km$^2$ in the Lhasa metropolitan area, it is recommended to implement differentiated land use policies in accordance with the requirements of establishing main functional zones based on the combined consideration of the future population distribution, economic and industrial layout, and land development patterns of the Lhasa metropolitan area. First of all, it is required to give great impetus to optimizing land use transition downtown. To prevent damage to regional resources and the environment from an excessive proportion of construction land, efforts should be made to strictly control the expansion of the scale of construction land in Chengguan District, particularly urban industrial and mining land. Secondly, it is required to effectively guarantee the land demand of population and economy in the central areas of surrounding cities and counties. Construction land supply should be appropriately increased in Dagze District, Doilungdeqen District, Gongga County, Nedong County, and other central districts and counties, and support for infrastructure construction should be strengthened to improve the overall benefits of land use. Thirdly, it is necessary to fully utilize the basic barrier role of ecological protection areas, strictly protect ecological land, and promote the harmonious development of the regional population, resources, and environment. The unique ecological environment and natural endowment is a crucial guarantee for the sustainable economic and social development of the Lhasa metropolitan area. In order to prevent any change in the use of ecological land, strict land use control is necessary for ecological function areas and land degradation control zones in the metropolitan region. Additionally, funding should be provided for ecological construction projects in the area for better ecological environment restoration.

(2)    Cross-regional division of cooperation should be facilitated.

Regions are closely linked in terms of resource sharing, market interconnection, facility interconnection, and environmental co-insurance, so reducing carbon emissions requires cross-regional division of labor and collaboration based on land use changes. The economic and social development and land use in downtown Lhasa have a significant impact on the surrounding areas. In essence, the surrounding areas' economic and social development, the land use, and especially the ecological and environmental conditions, have a profound impact on the downtown area of Lhasa. First of all, land use should be designated with the intention of maximizing each region's comparative advantages while facilitating a reasonable division of labor and interregional collaboration for overall planning and coordination

of land use. Second, increased efforts should be made to integrate construction land; direct an appropriate agglomeration of population and industry; promote the division of labor, cooperation, coordination, and complementarity among cities in the region; and create an urban land use pattern with a reasonable grade scale, accessible transportation, and a separation of primarily cultivated land from ecological functional areas [38]. Third, in order to effectively regulate and prevent duplicate construction and industrial structure convergence, rules and mechanisms for regulating land use in the Lhasa metropolitan area should be developed in coordination with other methods of regulating regional land use.

(3) More intensive land use should be strengthened.

Carbon emissions are greatly affected by the change in construction land, and the compact distribution of construction land will significantly inhibit the growth of carbon emissions. In addition, intensive cultivated land management and production will also inhibit the growth of carbon emissions. In the past, due to incomplete conversion of cultivated land and rapid advancement of construction land, the government's land development according to the regional master plan would result in fragmentation of land development, and this fragmented land use and development would increase the complexity within the region, while significantly contributing to the growth of local carbon emissions. As a result, the Lhasa metropolitan area should increase intensive land use to reduce fragmentation zones and reduce land-use carbon emissions. First of all, it should rationally arrange downtown construction land in planning, increase urban agglomeration, stimulate the development of surrounding areas, and promote healthy and rapid industrialization and urbanization [39]. Secondly, in addition to making efforts to increase the land area of the forest network and the water surface, it should strictly protect arable land, strengthen the protection and construction of concentrated and continuous high-standard basic cultivated land in the region, and promote the development of agriculture toward ecology, refinement, industrialization, and modernization. Thirdly, land consolidation in the inner city, particularly in the older urban areas, should be carried out to promote "urban renewal" and "stock development". It should focus on mitigating fragmentation during the period of land planning and development, ensuring consistency of laws between patches, and strengthening government enforcement to standardize the development [40].

We suggest that more research should be conducted to improve the precision of carbon emission and sink estimates in this area and to examine other socioeconomic factors associated with carbon emissions. To balance economic development and ecological preservation, more focus should also be placed on effective land use and urban planning.

**Author Contributions:** Original Draft, Methodology, Software, Visualization, D.K. and M.W.; Conceptualization, M.W., J.M. and W.M.; Supervision and Visualization, R.A. and M.W. All authors have read and agreed to the published version of the manuscript.

**Funding:** This work was supported by the National Natural Science Foundation of China (42201198), The Youth Science and Technology Fund of Gansu province (22JR5RA518); The Philosophy and social science Planning Project of Lanzhou City (22-A06) and The Central University Basic Research Fund of China of Lanzhou University (lzujbky-2022-46).

**Institutional Review Board Statement:** Not applicable.

**Informed Consent Statement:** Informed consent was obtained from all subjects involved in the study.

**Data Availability Statement:** Not applicable.

**Conflicts of Interest:** The authors declare no conflict of interest.

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
