# Peer review of "The Impacts of Land Use Spatial Form Changes on Carbon Emissions in Qinghai–Tibet Plateau from 2000 to 2020: A Case Study of the Lhasa Metropolitan Area"

_land, doi:10.3390/land12010122_

Round 1
Reviewer 1 Report
1- Methods and tools, which are essential to describe the work done by the authors, are not properly described.
2- It would be great if the authors could compare the results with other researchers' work if any. This valuable information could help other researchers which working in this field of study.
3- Besides, it would be great if the authors add more information about the drawbacks of the utilized methods if any.
Author Response
Dear Editors,
Thank you for your useful comments, questions and suggestions on the contents and structure of my manuscript. We have modified the manuscript accordingly, and the detailed corrections are listed below point by point:
Reviewer 1’s comments:
- Methods and tools, which are essential to describe the work done by the authors, are not properly described’
Response: Thanks so much for your valuable comments and suggestions. (1) The structure of "Materials and Methods" in Section 2 of the original draft is optimized, it showed as "2.1. Study area", "2.2. Data sources", "2.3. methods"; (2) "Selection of indexes" in 4.1 is combined in "2.3. Methods", and we added the introduction and comparison of SLM and SEM methods; (3) the part of "'5. Policy recommendations and conclusion' includes two sections in the new draft.
- It would be great if the authors could compare the results with other researchers' work if any. This valuable information could help other researchers which working in this field of study.
Response: Thanks so much for your valuable comments and suggestions. We've optimized the introduction.
- Besides, it would be great if the authors add more information about the drawbacks of the utilized methods if any. )
Response: Thanks so much for your valuable comments and suggestions. We modify the "Methods" section to include descriptions of SEM and SLM.
The authors is greatly thankful to editors and anonymous reviewers for sharing their valuable comments that significantly improved the quality of the paper. If you have any questions, please feel free to contact us. We appreciate your support very much.
Yours sincerely,
Meimei Wang
December 18th, 2022

Reviewer 2 Report
The authors explored the impact of land use on carbon emissions from the perspective of temporary and spatial characteristic. The research is innovative and significant. However, there have some problems need to be revised, the normalization of this manuscript should be improved.
1.The format of references citation should be revised. The authors should refer to the regulation of《Land》strictly.
2.The logic is a little mess. In line 38-39, the authors illustrated that the regional carbon emissions research based on land use changes is mainly focused on four areas. Firstly, the four aspects is distract. Secondly, the second aspect is only relate to carbon emission, it is not relate to land use. Thirdly, the third aspect is similar to fourth aspect, both of them are not relate to land use closely. Furthermore, what's the diffierence of urban form and urban spatial structure. I suggest the authors recategorize the several aspects. Fourthly, some sentence are repeat in different paragraph. Such as line 118-120 and 152-154. In addition, the authors should illustrate the impact of land use on carbon emission and the influence of carbon emission on ecological security specificly to show the importance of study area and the significant of this work. Fifthly, the similar sentences should be integrated. such as there have many sentences are emphasize the significance of this research.(line 120-122,line 129-133,line 140-142,line146-149). Sixthly, the sentence in line 123-125 is more suitable as a background content, it should be moved to the first paragraph in introduction.
3.In introduction, the authors should card the literature review clearly. The references are listed simply in this section at present, it is lack of summarizing. The authors' name of references are not suggest to emerge in text.
4.The sentences should explained more detailed. For example, the authors suppose the land use and carbon emissions of the Lhasa metropolitan area also show unique characteristics, it is different with that in midwest vally cities and eastern plains cities in China in line 157-159. What is the specific difference between the Lhasa metropolitan area and midwest vally cities, eastern plains cities? In addition, the authors choose this study area because of economic development and population in line 159-161. What's the relationship of land use ,carbon emissions with development and population?
5.The English need to be polished by native speaker. For instance, what's the meaning of land use changes and changes in the spatial patterns of construction land.
6.The manuscript contain forest land, grassland, cultivated land and so on. However, there only have construction land mentioned in section 1, section 2.1 and section 2.2.
7.In results and discussions, the structure and content should be adjusted as follow. The relationship between land form and carbon emission should be integrated into 2.2. The SLM and SEM should be supplied in methods. The section 3.1 should be integrated with 3.2. The section 3.3, 3.4 and 3.5 should be integrated into one part.
1.Introduction
2.Data source and Methods
2.1 Study area
2.2 Data sources
2.3 Methods
2.4 Relationship between land use and carbon emission
3.Results
3.1 Spatio-temporal evolution characteristics of land use
3.2 Spatio-temporal evolution characteristics of carbon emission on land use
4.Conclusions and policy recommendations
4.1 Conclusions
4.2 Policy recommendations
8.The results should emphasize the tempoary and spatial characteristic.
9.The text is not coorespond to the figures. For instance, the grassland has the widest distribution in line 216 in section 3.2, it didn't show the spatial of grassland in section 3.2. The characteristic of one core with many points in line 222-223 in section 3.2 is coorespond to figure 2 in section 3.1.
10. The rivers mentioned in line 235,236 should be added in figures.
11. In table 4 and table 5, the symbol of significant * should be supplied.
12.The conclusions are simple and distract. It should be integrated into three or four points.
13.Some relevant references are recommened. The authors should cite and refer to the structure and analysis of these references.
Spatial and temporal heterogeneity of urban land area and PM2.5 concentration in China. Urban Climate,2022,45:101268. doi:https://doi.org/10.1016/j.uclim.2022.101268.
Response characteristics and influencing factors of carbon emissions and land surface temperature in Guangdong Province,China.Urban Climate,2022,46:101330.doi:https://doi.org/10.1016/j.uclim.2022.101330.
Beating the urban heat:Situation,background,impacts and the way forward in China. Renewable and Sustainable Energy Reviews,2022,161,112350. doi:https://doi.org/10.1016/j.rser.2022.112350.
Author Response
Dear Editors,
Thank you for your useful comments, questions and suggestions on the contents and structure of my manuscript. We have modified the manuscript accordingly, and the detailed corrections are listed below point by point:
Reviewer 2’s comments:
The authors explored the impact of land use on carbon emissions from the perspective of temporary and spatial characteristic. The research is innovative and significant. However, there have some problems need to be revised, the normalization of this manuscript should be improved.
- The format of references citation should be revised. The authors should refer to the regulation of《Land》
Response: Thanks so much for your valuable comments and suggestions. The reference format of the manuscript has been modified in accordance with the《Land》reference format.
- The logic is a little mess. In line 38-39, the authors illustrated that the regional carbon emissions research based on land use changes is mainly focused on four areas. Firstly, the four aspects is distract. Secondly, the second aspect is only relate to carbon emission, it is not relate to land use. Thirdly, the third aspect is similar to fourth aspect, both of them are not relate to land use closely. Furthermore, what's the diffierence of urban form and urban spatial structure. I suggest the authors recategorize the several aspects. Fourthly, some sentence are repeat in different paragraph. Such as line 118-120 and 152-154. In addition, the authors should illustrate the impact of land use on carbon emission and the influence of carbon emission on ecological security specificly to show the importance of study area and the significant of this work. Fifthly, the similar sentences should be integrated. such as there have many sentences are emphasize the significance of this research.(line 120-122,line 129-133,line 140-142,line146-149). Sixthly, the sentence in line 123-125 is more suitable as a background content, it should be moved to the first paragraph in introduction.
Response: Thanks so much for your valuable comments and suggestions. (1) We delete the duplicate expression in lines 152-153 and modify it to"The Lhasa metropolitan area is affected by the plateau monsoon semi-arid climate and subsidence airflow and has bright weather throughout the year, with more than 3,000 hours of sunshine throughout the year." ; (2) The expressions of the impact of land use on carbon emissions and the impact of carbon emissions on ecological security are added at lines 35 and 38,"Carbon emissions from land use change are the most uncertain part of the global carbon cycle, and different types of land use can have an impact on carbon emissions due to their different land use intensities, human activities, and vegetation. For example, deforestation and other land cover changes typically release carbon from the terrestrial biosphere into the atmosphere in the form of CO2 (carbon dioxide), which can lead to significant increases in regional carbon emissions", "If countries do not take aggressive action, global average temperatures will eventually rise by 3-6 degrees Celsius. Climate change of this magnitude will likely cause sea levels to rise by tens of meters in the future. This catastrophe will lead to the extinction of almost half of the planet's species. ";(3) Lines 123-125 are moved to the first paragraph of the introduction "Also land urbanization has a significant impact on carbon emissions , and carbon emissions are the largest carbon source in the region, accounting for approximately 80% of the total regional carbon emissions."
- In introduction, the authors should card the literature review clearly. The references are listed simply in this section at present, it is lack of summarizing. The authors' name of references are not suggest to emerge in text.
Response: Thanks so much for your valuable comments and suggestions. We've optimized the introduction.
- The sentences should explained more detailed. For example, the authors suppose the land use and carbon emissions of the Lhasa metropolitan area also show unique characteristics, it is different with that in midwest vally cities and eastern plains cities in China in line 157-159. What is the specific difference between the Lhasa metropolitan area and midwest vally cities, eastern plains cities? In addition, the authors choose this study area because of economic development and population in line 159-161. What's the relationship of land use ,carbon emissions with development and population?
Response: Thanks so much for your valuable comments and suggestions.In the part of "3.1. Spatio-temporal evolution characteristics of land use",we analyzed the differences in land use and carbon emissions between Lhasa and other cities. In" 2.5. Impact logic of land use form on carbon emission",we discussed the relationship between land use, carbon emissions and development and population.
- The English need to be polished by native speaker. For instance, what's the meaning of land use changes and changes in the spatial patterns of construction land.
Response: Thanks so much for your valuable comments and suggestions. We have asked an English expert with Geography knowledge help us to improve the expression of the manuscript .
- The manuscript contain forest land, grassland, cultivated land and so on. However, there only have construction land mentioned in section 1, section 2.1 and section 2.2.
Response: Thanks so much for your valuable comments and suggestions. The correlation analysis results in Figure 5 show that changing the land form of forest land, grassland, water space, unused land and cultivation land have no significant effect on carbon emissions and carbon source/sink, whereas changing of construction land has a significant effect on carbon emissions. Then, we will analyze the impact of construction land form on carbon emissions of land use.
- In results and discussions, the structure and content should be adjusted as follow. The relationship between land form and carbon emission should be integrated into 2.2. The SLM and SEM should be supplied in methods. The section 3.1 should be integrated with 3.2. The section 3.3, 3.4 and 3.5 should be integrated into one part.
Response: Thanks so much for your valuable comments and suggestions. (1) The structure of "Materials and Methods" in Section 2 of the original draft is optimized, it showed as "2.1. Study area", "2.2. Data sources", "2.3. methods"; (2) "Selection of indexes" in 4.1 is combined in "2.3. Methods", and we added the introduction and comparison of SLM and SEM methods; (3) the part of "'5. Policy recommendations and conclusion' includes two sections in the new draft.
- The results should emphasize the tempoary and spatial characteristic.
Response: Thanks so much for your valuable comments and suggestions. In the part of "3.1. Spatio-temporal evolution characteristics of land use",we add in changes in land use over time and space.
- The text is not coorespond to the figures. For instance, the grassland has the widest distribution in line 216 in section 3.2, it didn't show the spatial of grassland in section 3.2. The characteristic of one core with many points in line 222-223 in section 3.2 is coorespond to figure 2 in section 3.1.
Response: Thanks so much for your valuable comments and suggestions. We adjusted the description to match the picture one by one. then we merge original articles 3.1 and 3.2 into "Spatio-temporal evolution characteristics of land use", we merge 3.3, 3.4, 3.5 to "3.2 Spatio-temporal evolution characteristics of carbon emission on land use".
10.The rivers mentioned in line 235,236 should be added in figures.
Response: Thanks so much for your valuable comments and suggestions. We remade Figure 4.
- In table 4 and table 5, the symbol of significant * should be supplied.
Response: Thanks so much for your valuable comments and suggestions. the symbol of significant * are supplied in Table 3 and Table 4.
12.The conclusions are simple and distract. It should be integrated into three or four points.
Response: Thanks so much for your valuable comments and suggestions. We rewrote the conclusion of the paper in "5.1. Conclusion".
- Some relevant references are recommened. The authors should cite and refer to the structure and analysis of these references.
Response: Thanks so much for your valuable comments and suggestions. We have adopted the literature you recommended, and we have also added some new literature.
The authors is greatly thankful to editors and anonymous reviewers for sharing their valuable comments that significantly improved the quality of the paper. If you have any questions, please feel free to contact us. We appreciate your support very much.
Yours sincerely,
Meimei Wang
December 18th, 2022

Round 2
Reviewer 1 Report
The authors have substantially revised the original submission so I recommend this work can be considered for publication
Author Response
The authors is greatly thankful to editors and anonymous reviewers for sharing their valuable comments that significantly improved the quality of the paper. We appreciate your support very much.
Yours sincerely,
December 27th, 2022
Reviewer 2 Report
The logic of structure of this manuscript have improved. However, there still have some problems need to be revised.
1.In abstract, why the authors use many semicolon rather than comma.
2.In introduction, the logic of context should be improved, especially in line 72-97.
3.In line 73, the 2 should be subscripted rather than superscripted.
4.In line 74-75, there have national,provincial and local three scale, it should correspond to three references. However, this sentence only show two references.
5.In line 83, what's the meaning of the second is on the methods and methods?
6.The figure of study area should be supplied in section 2.1.
7.In 2.3, the format of several sentences is not united.
8.In line 134-135, the authors suppose the land use and carbon emissions of the Lhasa metropolitan area also show unique characteristics, it is different with that in midwest valley cities and eastern plains cities in China. What‘s the specific of unique characteristic?
9.In section 3 results and discussion, the discussions often integrate with conclusions, rather than results.
10. In line 263, it is lack of a half bracket. In line 271, there should have a space between to and 2015.
Author Response
Explanation of the revision for [Land] Manuscript ID: land-2106270
Dear Editors,
Thank you for your useful comments, questions and suggestions on the contents and structure of my manuscript. We have modified the manuscript accordingly, and the detailed corrections are listed below point by point:
The logic of structure of this manuscript have improved. However, there still have some problems need to be revised.
- In abstract, why the authors use many semicolon rather than comma.
Response: Thanks so much for your valuable comments and suggestions. We have changed the semicolon in the abstract to comma.
- In introduction, the logic of context should be improved, especially in line 72-97.
Response: Thanks so much for your valuable comments and suggestions. The logic of context in line 72-97 are re-optimized.
- In line 73, the 2 should be subscripted rather than superscripted.
Response: Thanks so much for your valuable comments and suggestions. In line 73, we change '2' to a subscript.
- In line 74-75, there have national,provincial and local three scale, it should correspond to three references. However, this sentence only show two references.
Response: Thanks so much for your valuable comments and suggestions. According to the second suggestion of the reviewer, the logic of lines 72-97 was improved: (1) the original 'the carbon emission estimation model' was expanded to 'using the carbon emission estimation model to estimate the carbon emission intensity and carbon footprint of different industrial spaces in China'; (2) Move 91 lines', finding that urban form influnces internal carbon emissions and believing that it plays an important role in reducing carbon emissions (Khan&Pinter, 2016; Shi et al., 2020) 'to Part 4' The fourth is on the impact of urban spatial structure on carbon emissions'; (3) In line 89-90, 'Many researchers studied agricultural land transformation (Wang et al., 2018), urban heat island effect (Palme et al., 2017), and urban households (Ding&Li, 2017)' is revised to 'Many researchers studied the impact of agricultural land transformation (Wang et al., 2018), urban heat island effect (Palme et al., 2017), And urban households (Ding&Li, 2017) on carbon emissions', to increase the logic of text.
- In line 83, what's the meaning of the second is on the methods and methods?
Response: Thanks so much for your valuable comments and suggestions. We modify "second is on the methods and methods" in line 83 to "The second is the different research methods of carbon emission. drivers'.
- The figure of study area should be supplied in section 2.1.
Response: Thanks so much for your valuable comments and suggestions. We have provided photos of the study area in Section 2.1.
- In 2.3, the format of several sentences is not united.
Response: Thanks so much for your valuable comments and suggestions. We have unified the sentence format in paragraph 2.3.
- In line 134-135, the authors suppose the land use and carbon emissions of the Lhasa metropolitan area also show unique characteristics, it is different with that in midwest valley cities and eastern plains cities in China. What‘s the specific of unique characteristic?
Response: Thanks so much for your valuable comments and suggestions. The spatial distribution of carbon emissions from land use in the Lhasa metropolitan area is different from that in the eastern cities. Carbon emissions in Cities in eastern China, such as the Pearl River Delta, have a centralized carbon emission core, which is larger in scope and gradually expands outward from the periphery (Xu et al,2018). The construction land and agricultural land in the Lhasa metropolitan area form a carbon emission pattern of "one core and multiple belts" with a very small carbon emission core area. The "multi-belt" carbon emission distribution formed by agricultural land is due to the lack of heat in the Qinghai-Tibet region and the relatively low annual effective temperature, which cannot meet the maturation requirements of general crops. Therefore, in order to solve the problem of heat deficiency, the plantation industry in Qinghai-Tibet region is concentrated in the valleys of Yarlung Tsangpo River and Huangshui River and its tributaries, forming a unique "one core and multiple belts" carbon emission pattern. See line 335.
- In section 3 results and discussion, the discussions often integrate with conclusions, rather than results.
Response: Thanks so much for your valuable comments and suggestions. We amend Section 3 "Results and discussion" to "Results and Analysis".
- In line 263, it is lack of a half bracket. In line 271, there should have a space between to and 2015.
Response: Thanks so much for your valuable comments and suggestions. We add a half-parenthesis to line 263, 'the average carbon emission intensity for construction land is 3.607 kg/(m2·a) (Yan et al., 2010)', line 271, add a space between to and 2015.
The authors is greatly thankful to editors and anonymous reviewers for sharing their valuable comments that significantly improved the quality of the paper. If you have any questions, please feel free to contact us. We appreciate your support very much.
Yours sincerely,
December 27th, 2022
